# A Monte Carlo Study of Hyperon Production with the MPD and BM@N Experiments at NICA

Alexander Zinchenko [1,*], Mikhail Kapishin [1], Viktar Kireyeu [1], Vadim Kolesnikov [1], Alexander Mudrokh [1], Dilyana Suvarieva [1,2], Veronika Vasendina [1] and Dmitry Zinchenko [1]

1   Veksler and Baldin Laboratory of High Energy Physics, Joint Institute for Nuclear Research, 6 Joliot-Curie, Dubna 141980, Russia
2   Department of Physics, Plovdiv University, Car Assen, 24, BG-4000 Plovdiv, Bulgaria
*   Correspondence: alexander.zinchenko@jinr.ru

**Abstract:** Study of the strangeness production in heavy-ion collisions is one of the most important parts of the physics program of the BM@N and MPD experiments at the NICA accelerator complex. With collision energies $\sqrt{s_{NN}}$ of 2.3–3.3 GeV in the fixed target mode at BM@N and 4–11 GeV in the collider mode at MPD, the experiments will cover the region of the maximum net baryon density and provide high-statistics complementary data on different physics probes. In this paper, some results of Monte Carlo studies of hyperon production with the BM@N and MPD experiments are presented, demonstrating their performance for investigation of the objects with strangeness.

**Keywords:** heavy-ion collisions; strangeness production; hyperon reconstruction; NICA complex; MPD experiment; BM@N experiment

## 1. Introduction

The experimental exploration of the high-density nuclear matter is an important subject for present and future research with heavy-ion beams. This is also one of the main research directions of the Nuclotron-based Ion Collider Facility (NICA) which is currently under construction at the Joint Institute for Nuclear Research (JINR) in Russia [1]. The accelerator complex will provide unique possibilities to produce baryonic matter at high densities. Two experimental setups are intended to study properties of this state of nuclear matter. The Multi-Purpose Detector (MPD) at the NICA collider [2] was designed to measure different physics processes and is being assembled at present to provide new information on the quantum chromodynamic (QCD) phase diagram at large baryon-chemical potentials, including the high-density equation of state (EOS). Another experimental setup, Baryonic Matter at Nuclotron (BM@N) [3], has been upgraded to its full configuration and carried out its first run in December 2022–January 2023 with Xe beam on CsI target. The two experiments will provide complete coverage of the region of the maximum net baryon density [4] (Figure 1).

Strangeness production is of particular interest at energies available at NICA [5]. New experimental data in this energy region could help to understand the origin of an enhancement of the strangeness production in heavy-ion collisions which was experimentally observed at SPS [6] and RHIC [7] and can be explained at present by different model scenarios [8,9]. Precise measurements of yields of strange hadrons may help to better understand strangeness production mechanisms in nuclear collisions [10–13] and to improve constraints on the chemical freeze-out parameters. Study of subthreshold production of multistrange hyperons can serve as a powerful tool to determine the high-density equation of state of baryonic matter [14].

In order to better understand the dynamics of hot and dense hadronic matter, in particular, the strangeness production mechanism, the MPD and BM@N experiments at the NICA accelerator complex will provide new precise data on the total yields, rapidity,

transverse momentum, and azimuthal angle distributions of strange particles, including (anti)-hyperons and hypernuclei. In this paper, some results of Monte Carlo simulations of these experiments are presented, showing their ability to reconstruct hyperons and study strangeness production in heavy-ion interactions.

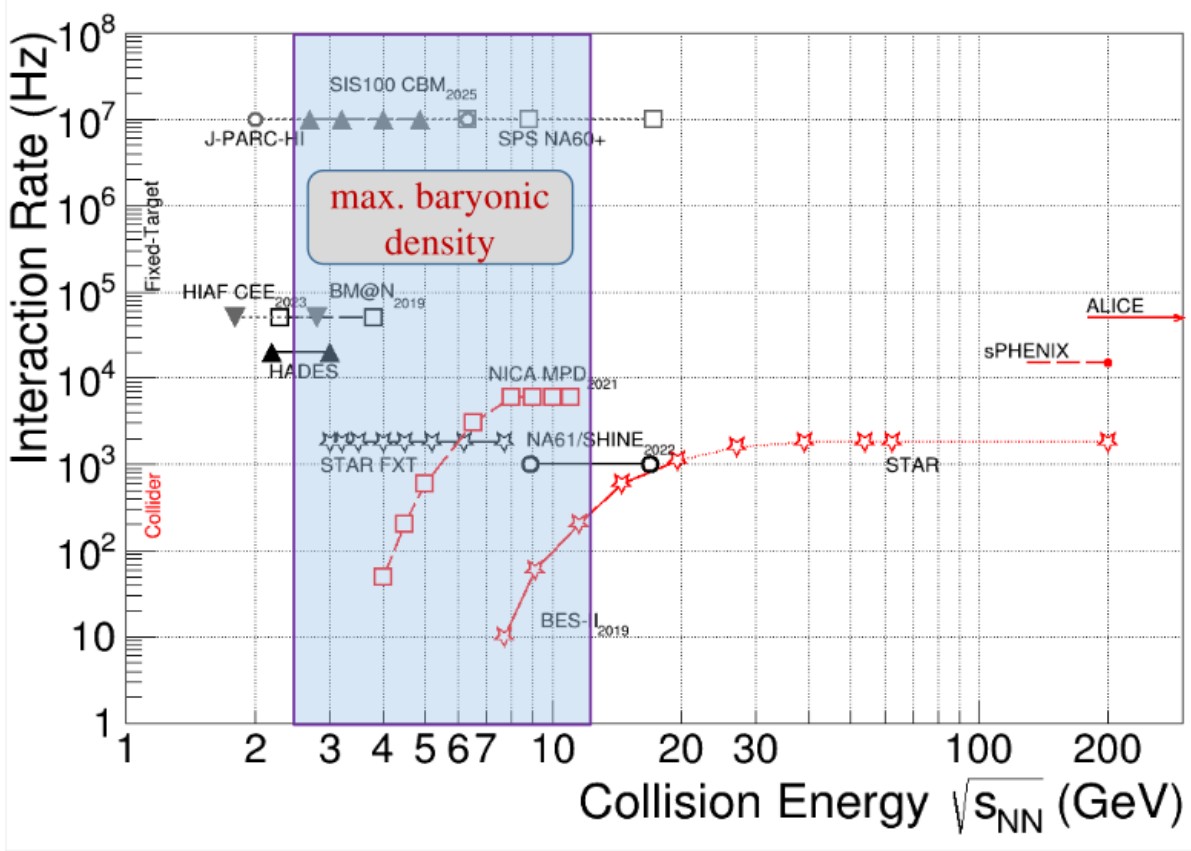

**Figure 1.** Landscape of heavy-ion experiments.

## 2. MPD and BM@N Detectors

The Multi-Purpose Detector [15] at the NICA collider was designed to detect hadrons, electrons, and photons over a large phase-space with the Au + Au collision rate of ∼7 kHz at the designed luminosity of $L = 10^{27}$ cm$^{-2}$s$^{-1}$.

Figure 2 shows a three-dimensional view of the MPD setup. It has a configuration typical for collider detectors. The momentum measurement will be carried out in a magnetic field with a nominal strength of 0.5 T directed along the beam axis. The field is produced by a superconducting solenoid which will host all subdetectors. The MPD main tracking detector TPC (time-projection chamber) offers an effective pseudorapidity coverage of $|\eta| < 1.5$, where it allows us to perform the reconstruction of charged particle trajectories and to determine momenta with an accuracy better than 3.5% at transverse momenta $p_T$ below 2 GeV/$c$ [16]. The TPC also provides particle identification via the specific energy loss measurement ($dE/dx$) in the TPC gas with a resolution better than 8% (as was also demonstrated by measurements in the detector of a similar design at the STAR experiment [17]). Identification of charged particles in the intermediate momentum region, where hadrons can not be discriminated by $dE/dx$ measurements in the TPC, will be guaranteed by the the time-of-flight (TOF) system, surrounding the TPC, and composed of the multi-gap resistive plate chamber (MRPC). The overall TOF time resolution of about 80 ps will provide a $\pi/K$ separation better than $3\sigma$ up to 1.2 GeV/$c$ and $K/p$ separation up to 2.5 GeV/$c$. The electromagnetic calorimeter (ECAL), with a projective geometry, composed of modules of the "shashlyk" lead-scintillator type [18], will measure

the spatial position and energy of photons as well as improve identification of electrons in the high-multiplicity environment of the experiment. The MPD forward detector (FD), consisting of two arrays of quartz Cherenkov detectors for detection of high-energy photons and relativistic charged particles, will provide fast timing and trigger with a 40 ps time resolution. Two arms of the forward hadron calorimeters (FHCALs), composed of 44 lead-scintillator modules each, cover the pseudorapidity range $2.8 < |\eta| < 4.5$. The FHCAL will be used for event characterization, i.e., determination of the centrality and event plane orientation from the measurement of the forward-going energy distribution. More details on the MPD detector can be found in Ref. [19].

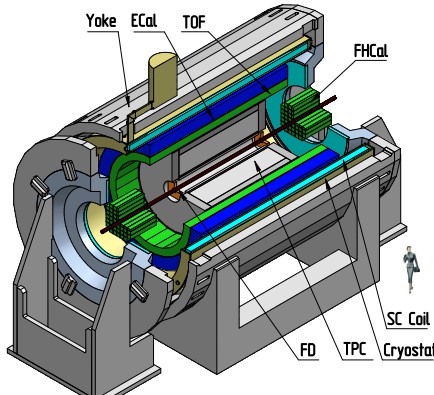

**Figure 2.** Three-dimensional view of the MPD detector.

The current layout of the BM@N setup is presented in Figure 3. Tracks of charged particles produced in the interaction are reconstructed with the hybrid tracking system consisting of four stations of double-sided microstrip silicon modules (forward silicon) downstream from the target, and a set of seven stations of two-coordinate GEM (gaseous electron multiplier) detectors with strip readout mounted downstream from the silicon tracker. The forward silicon contains 6, 10, 14, and 18 double-sided silicon modules in stations 1–4, respectively, arranged into pairs of half-stations below and above the beam line at a distance of ∼10 cm between stations. Each module with a width of 6 and a height of 12 cm has a strip pitch of 100 μm and a stereo angle of 2.5°. The GEM stations with a strip pitch of 800 μm and a stereo angle of 15° have a size of ± ∼80 and ± ∼40 cm in horizontal and vertical directions, respectively, and are arranged at a distance of ∼30 cm from each other along the beam (Z-axis). The forward silicon tracker and the GEM stations are positioned inside a large aperture dipole magnet with a gap height of 1 m, providing a vertical (along Y-axis) magnetic field with a maximum value of 1.0 T. Particle identification is provided by the time-of-flight system consisting of two walls of resistive plate chambers (mRPC) with strip readout, located at ∼4 m (TOF400 subsystem) and ∼7 m (TOF700 subsystem) downstream from the target. The cathode strip chambers (CSCs) improve the quality of matching of reconstructed tracks with TOF measurements. Electromagnetic probes are measured with the electromagnetic calorimeter ECAL, and the interaction centrality and event plane are determined by the zero-degree calorimeter ZDC. The interaction trigger is produced with the BD (barrel detector) and SiD (silicon detector) detectors, covering low- and high-rapidity regions, respectively. More details on the BM@N detector systems can be found in Ref. [20]. It should be noted here that in the last few years, the experiment performed several technical runs with light ion beams in partial detector configuration [21–23]. The results obtained proved the detector concept and demonstrated validity of the data processing and event reconstruction approaches.

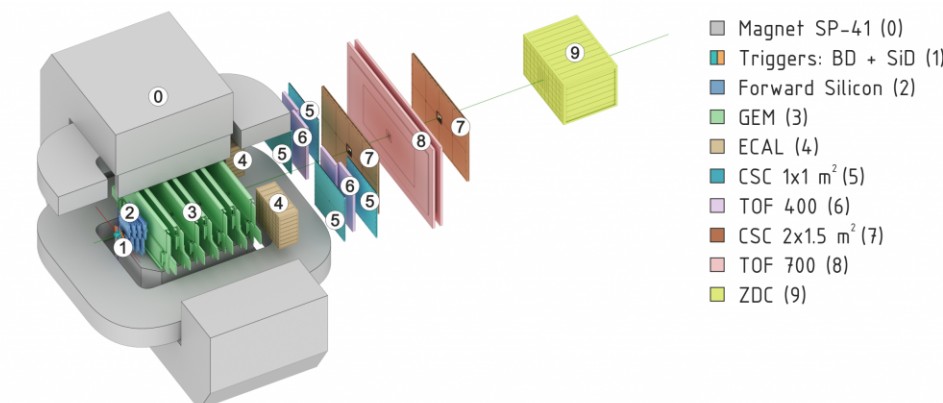

**Figure 3.** Three-dimensional view of the BM@N detector. The ion beam comes from the left through the vacuum beam pipe to avoid interactions with the air (not shown).

## 3. Hyperon Reconstruction

Hyperons are identified via their weak decays to charged particles in the final state. Such decays produce very distinct event topologies, as can be seen in Figure 4, where schematic views of the decay topology of $\Lambda$ and $\Omega$ hyperons in the bending plane of the magnetic field, i.e., in the plane perpendicular to the magnetic field, are shown. Such topologies can be selected by making use of the secondary vertex reconstruction method, which is based on finding secondary (decay) vertices separated from the primary one (interaction point). The selection is checked by applying different selection criteria on the relevant kinematic and topological variables. For example, track combinations are accepted only if the distance of the closest approach $dca_{V0}$ in space between the daughter track candidates is smaller than some value. This cut ensures that the tracks originate from the same mother particle. To select decays of long-lived particles, it is usually required that the secondary vertex position is located farther than some distance *path* from the primary one. In order to suppress the combinatorial background due to primary tracks, the minimum value of the decay track impact parameters to the primary vertex $dca_{K,p,\pi}$ should be greater than some threshold. The quality of the topology reconstruction can be also enhanced by requiring the impact parameter of the decayed particle with respect to the primary vertex or its pointing angle, defined as the angle between its momentum and the direction vector from the primary to the secondary vertex, to be smaller than some value. For selected particle combinations, the invariant mass is calculated under the corresponding daughter particle hypotheses, e.g., a proton and a pion for the case of V0 or $\Lambda$, and a kaon for $\Omega$ hyperon. Peaks in the invariant mass distributions at the right particle mass values serve as a clear signature of the hyperon decay under study.

The decay reconstruction package is built around vertex-fitting procedures based on the Kalman filtering technique [24–26] of the MpdRoot [27] and BmnRoot [28] software frameworks of the MPD and BM@N experiments, respectively.

The exact selections can be found by performing a multidimensional scan over the whole set of selection criteria with a requirement to maximize the invariant mass peak significance. The significance is defined as $S/\sqrt{S+B}$, where S and B are the total numbers of signal and background combinations in the mass window defined by the signal peak shape and width. For this study, a simple mass interval estimate is taken to be $\pm 2\sigma$ around the peak position, obtained from a fit of the invariant mass distribution to a sum of a Gaussian and polynomial function, with $\sigma$ being the Gaussian width parameter.

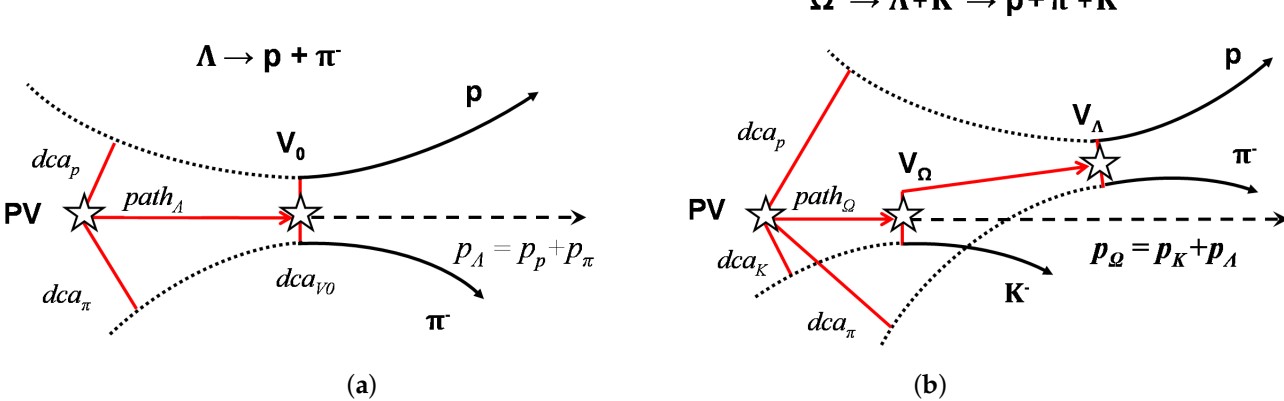

**Figure 4.** Views of hyperon decay topologies in the bending plane of the magnetic field: (**a**) two-prong decays of a neutral particle ("V0-decays") (e.g., $\Lambda \to p + \pi^-$); (**b**) cascade-type decays (e.g., $\Omega^- \to \Lambda + K^- \to p + \pi^- + K^-$). Here, $dca_p$, $dca_\pi$, and $dca_K$ are the distances of the closest approach (DCA) of the decay tracks to the primary vertex $PV$; $dca_{V0}$ is the distance between daughter tracks in the mother decay vertex $V_0$; *path* is the mother particle decay length; and $\mathbf{p_P}$, $\mathbf{p_\beta}$, $\mathbf{p_K}$, $\mathbf{p_\Lambda}$, and $\mathbf{p_\Omega}$ are momenta of particles.

To obtain more accurate numbers for hyperon yield extraction, one can use a background subtraction procedure based on the event mixing techniques, as can be found, for example, in [29]. The parameter scan was performed using nested loops over selection criteria in small steps, producing the invariant mass peak significance for each set of selection cuts. The final set of selection cuts was chosen based on the maximum significance value achieved. The parameter values obtained can be seen in Tables 1 and 2.

**Table 1.** Selection criteria used for $V^0$ ($\Lambda$ and $\overline{\Lambda}$) reconstruction. Cuts on DCAs are imposed in the $\chi^2$-space, i.e., after normalization to respective parameter errors.

| Cut | $\Lambda$ | $\overline{\Lambda}$ |
|---|---|---|
| DCA of daughters to primary vertex | $> 5.0(\pi), > 2.5(p)$ | $> 4.0(\pi), > 1.5(p)$ |
| DCA between daughters | <3.0 | <2.8 |
| Decay length, cm | >2.5 | >2.5 |
| Mother pointing angle, rad | <0.08 | <0.14 |

**Table 2.** Selection criteria used for cascade ($\Xi^-$ and $\Omega^-$) reconstruction. Cuts on DCAs are imposed in the $\chi^2$-space, i.e., after normalization to respective parameter errors.

| Cut | $\Xi^-$ | $\Omega^-$ |
|---|---|---|
| DCA of daughters to primary vertex | $> 6.5(\pi), > 2.0(\Lambda)$ | $> 6.0(K), > 3.0(\Lambda)$ |
| DCA between daughters | <2.5 | <1.5 |
| Decay length, cm | >2.0 | >1.5 |
| Mother pointing angle, rad | <0.08 | <0.10 |
| DCA of $\Lambda$ daughters to primary vertex | $> 6.0(\pi), > 3.8(p)$ | $> 6.5(\pi), > 3.5(p)$ |
| DCA between $\Lambda$ daughters | <3.5 | <2.8 |
| $\Lambda$ decay length, cm | >3.5 | >3.0 |
| $\Lambda$ pointing angle, rad | <0.20 | <0.22 |

The scanning procedure is quite time-consuming (especially for cascade decays) and cannot fully take into account nonlinear correlations between different selection parameters. However, at present, different multivariate approaches based on machine learning (ML)

techniques are available, and it was already demonstrated that they can improve the selection quality of short-lived particle decays [30]. One of such implementations, namely, the TMVA package [31] of the ROOT framework, was used to test the performance of the ML-based approach in this study. Figure 5 shows an example of the plots produced by the TMVA package, in particular, distributions of input variables for signal and background particle combinations during $\Lambda$ reconstruction and network output, which is used as a classification variable to separate the signal from the background after proper network training.

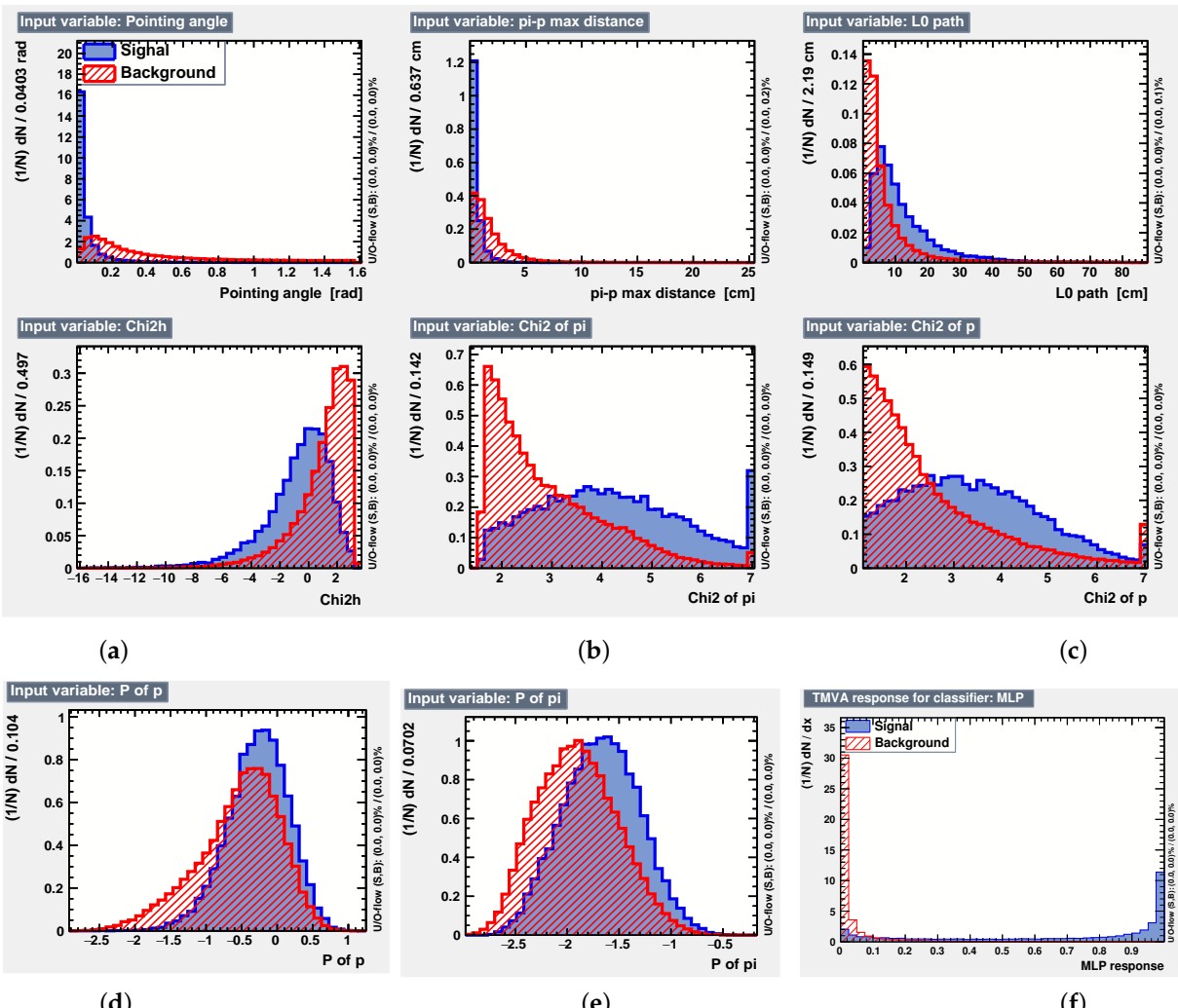

**Figure 5.** (**a**–**e**) Distributions of input variables for the TMVA package for the case of $\Lambda$ reconstruction: angle between hyperon momentum and the direction vector from the primary to the secondary vertex, $p - \pi^-$ distance in the decay vertex, hyperon decay length, $\log(\chi^2)$ of reconstructed secondary vertex, $\chi^2$ of $p$ and $\pi^-$ with respect to the primary vertex (*DCAs* normalized to their respective errors), $log(momentum)$ of $p$ and $\pi^-$; (**f**) TMVA output variable distribution when the multilayer perceptron (MLP) is used as a discriminating network. Blue and pink histograms represent signal and background combinations, respectively.

## 4. Results

### 4.1. MPD Setup

Basic performance numbers of the MPD experiment for track reconstruction and particle identification as obtained from the Monte Carlo simulation can be found in Ref. [19]. The simulation was performed with the GEANT4 particle transport package and included a realistic description of the detector response [16] within the MpdRoot framework.

Figure 6 shows some examples of the reconstructed invariant mass spectra of identified particle combinations for $\Lambda$, $\Xi^-$ and $\Omega^-$ hyperon decays, obtained after applying the selection criteria described above, for PHSD [32] simulated minimum bias Au+Au collisions at $\sqrt{s_{NN}} = 11$ GeV, produced with a primary vertex distributed according to a Gaussian with $\sigma$ of 25 cm along the beam line. One can see clear peaks with high significance and signal/background ratio, obtained with efficiencies allowing us to perform high-statistics study of hyperon properties. From a comparison of Figure 6a–f, it can also be seen that the results improve if the ML-based procedure configured and trained for each particular particle specie is used to suppress the background. The somewhat different shapes of the background for the two selection approaches also suggest that the ML-based method takes advantage of additional correlations of discrimination variables in comparison with the simple cut method, as hoped for. The acceptance of the MPD detector for hyperon studies can be evaluated from Figure 7, where the $y - p_T$ phase space of reconstructed hyperons is presented. One can see that the detector provides a rapidity coverage of $|y| \lesssim 1.2$ for $\Lambda$ hyperons with a high uniformity at mid-rapidity $|y| < 0.5$. The rapidity interval is defined by the TOF system particle identification, and can be extended up to $|y| \lesssim 1.5$ at the expense of somewhat worse identification capability via the TPC $dE/dx$ only. The low-$p_T$ threshold of reconstructed $\Lambda$ hyperons is about 0.2 GeV/$c$, as can be also seen in Figure 8, where the total reconstruction efficiency is plotted as a function of transverse momentum. The efficiency includes the decay branching ratio and detector acceptance, as well as reconstruction and selection efficiencies.

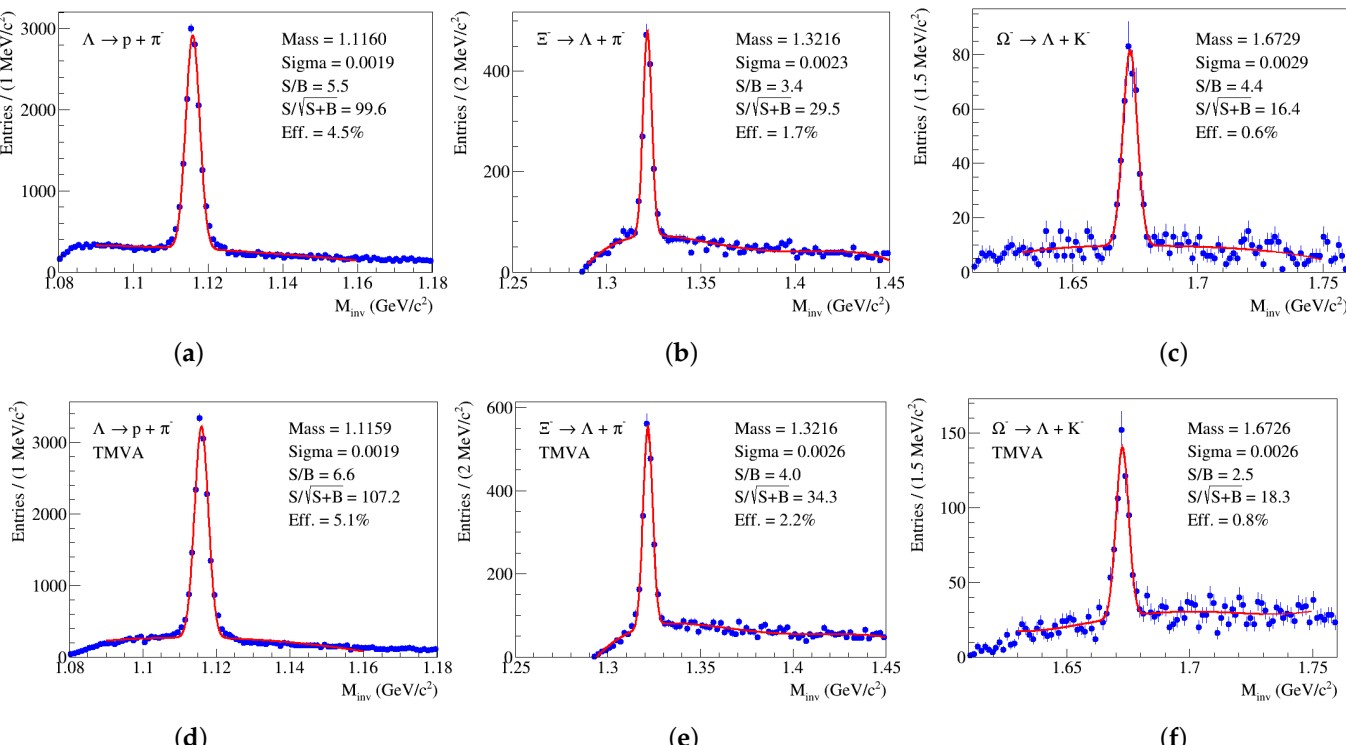

**Figure 6.** Reconstructed invariant mass spectra of hyperon decay products: (**a**) $\Lambda \to p + \pi^-$, (**b**) $\Xi^- \to \Lambda + \pi^-$, (**c**) $\Omega^- \to \Lambda + K^-$. Background suppression is based on the topological cut approach. Panels (**d**–**f**) are the same as (**a**–**c**) for the hyperon selection procedure based on machine learning techniques. The results are shown for different numbers of processed events: $2 \cdot 10^4$ for (**a**,**d**), $10^5$ for (**b**,**e**), $2 \cdot 10^6$ for (**c**,**f**). Histograms are fitted to a combination of a Gaussian and a polynomial function. The results obtained from the fits, i.e., mean and sigma of the Gaussian, signal/background ratio, significance, and signal reconstruction efficiency, are presented in the legends. The signal and background are calculated within a $\pm 2\sigma$ interval of the peak position, and efficiency is defined with respect to all hyperons produced within 50 cm from the collision point.

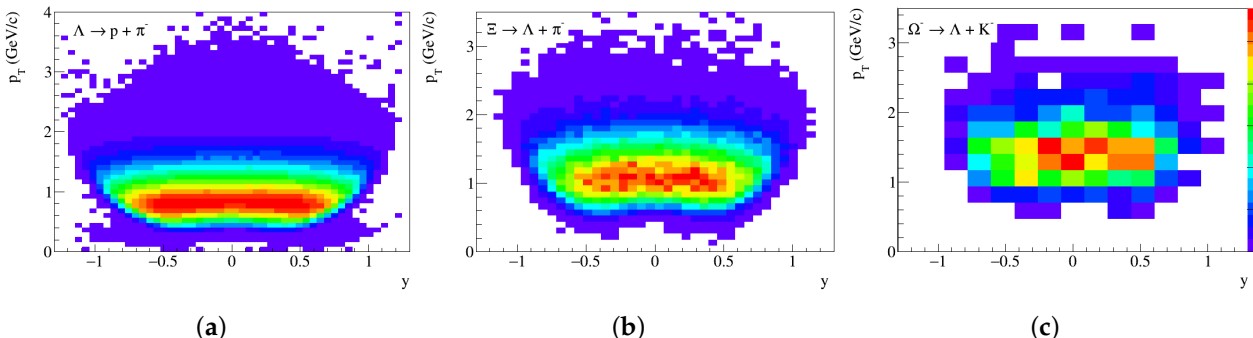

**Figure 7.** Transverse momentum $p_T$ vs. rapidity $y$ phase space of reconstructed true hyperons: (**a**) $\Lambda$, (**b**) $\Xi^-$, (**c**) $\Omega^-$.

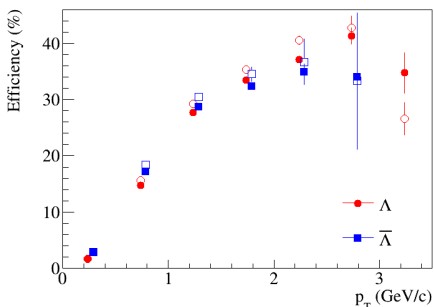

**Figure 8.** $\Lambda$ and $\overline{\Lambda}$ total reconstruction efficiency as a function of transverse momentum in central (impact parameter of the collisions $b < 3.3$ fm—filled symbols) and peripheral ($b = 9 - 13$ fm—empty symbols) interactions.

The reconstructed invariant mass spectra were used to determine hyperon multiplicities in different $p_T$-bins which, after correction for efficiency (Figure 8), allow producing some physics distributions such as invariant $p_T$-spectra Figure 9a,b. Fitting of such spectra to proper functions [29] can be used to correct for the missing parts at low and high $p_T$ values, which are estimated to be ~5–7% for $p_T$ below 0.2 GeV/$c$, depending on event centrality, and below 1% for high transverse momenta. Therefore, it allows us to obtain hyperon rapidity densities and total hyperon yields and extract, for example, antihyperon-to-hyperon yield ratio as a function of transverse momentum to study critical phenomena [33] (Figure 9c).

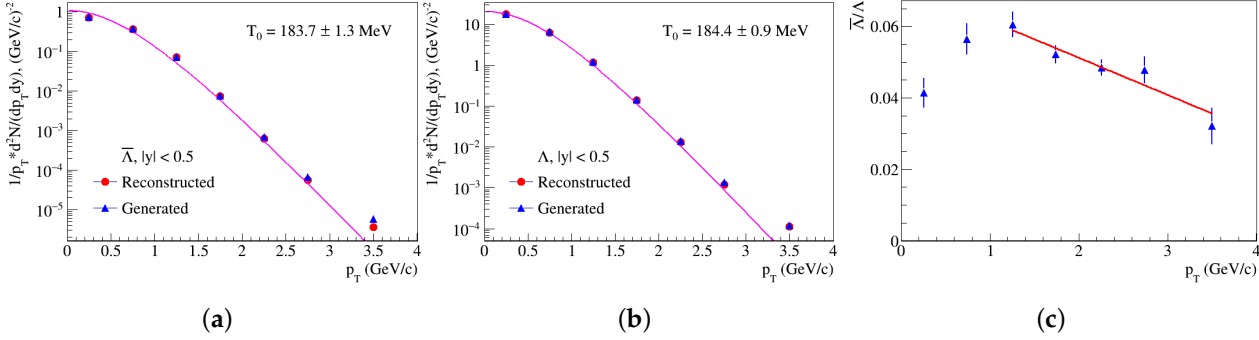

**Figure 9.** Invariant transverse momentum spectra of $\overline{\Lambda}$ (**a**) and $\Lambda$ (**b**) hyperons at mid-rapidity ($|y| < 0.5$). The reconstructed data are indicated by circles, while triangles represent the spectra from the model. The lines show the thermal fits of the form $1/p_T \cdot d^2N/dp_T/dy \sim \exp[-(\sqrt{p_T^2 + m^2} - m)/T_0]$ with effective temperature $T_0$. (**c**) Mid-rapidity $\overline{\Lambda}/\Lambda$-ratio in 0–5% central collisions as a function of $p_T$. The line indicates a fit to the linear function $a + b \cdot p_T$. Results are shown for $4 \cdot 10^7$ minimum bias Bi+Bi collisions at $\sqrt{s_{NN}} = 9$ GeV produced with the PHQMD generator [34].

*4.2. BM@N Setup*

Performance of the BM@N experiment with respect to track reconstruction and particle identification evaluated from GEANT4-based realistic Monte Carlo simulation within the BmnRoot software framework is presented in [35]. The results were obtained for different beam energies and corresponding values of the magnetic field adjusted to keep the same curvature of the beam particles inside the carbon fiber vacuum beam pipe. Since the TOF subdetectors decrease geometric acceptance of the setup, hyperon reconstruction in the present study is performed for daughter particles without identification, i.e., mass values are assigned to reconstructed tracks according to their charge and the decay mode studied.

Figure 10 shows invariant mass spectra of $\Lambda$-candidates reconstructed in Monte Carlo event samples of Xe+Cs interactions for Xe beam kinetic energies of 1.5 A and 2.9 A GeV together with the $y - p_T$-phase space distributions of simulated and reconstructed hyperons. The events are produced with the DCM-SMM generator [36]. The selection cuts applied are presented in Table 3. The difference with the MPD setup (Table 1) in the decay length and pointing angle mostly comes from the difference in the kinematics, i.e., higher Lorentz boost in the fixed target configuration. From Figure 10, one can see that the detector provides coverage of the positive part of the rapidity spectrum (with respect to the central value) as typical for fixed-target experiments, and sufficient coverage in $p_T$ with the low-$p_T$ threshold similar to that of the MPD setup (cf. Figure 7). For higher beam momentum, the invariant mass resolution improves because of better momentum reconstruction accuracy in the higher magnetic field, which (together with a higher production rate) results in a higher hyperon reconstruction efficiency. The resulting hyperon statistics for a moderate number of collected events will allow studying the excitation function of $\Lambda$ hyperon production. At the highest energy of 3.9 A GeV, the $\Xi^-$-hyperon peak becomes quite visible as well (Figure 11). Selection of $\Xi^-$-decays was performed using the TMVA package. It should be noted here that for relatively low interaction energies, the role of ML approaches will increase due to low particle production rate near threshold.

**Table 3.** Selection criteria used for $\Lambda$ reconstruction with BM@N at different beam energies. Cuts on DCAs are imposed in the $\chi^2$-space, i.e., after normalization to respective parameter errors.

| Cut | 1.5A GeV | 1.9 A GeV | 3.9 A GeV |
|---|---|---|---|
| DCA of daughters to primary vertex | $> 5.0(\pi), > 2.5(p)$ | $> 2.5(\pi), > 1.5(p)$ | $> 2.5(\pi), > 1.3(p)$ |
| DCA between daughters | <4.3 | <5.0 | <3.5 |
| Decay length, cm | >5.0 | >4.0 | >4.0 |
| Mother pointing angle, rad | <0.01 | <0.01 | <0.01 |

The simulation results obtained make it possible to evaluate the expected statistics of reconstructed hyperons in the first BM@N experimental run with the full configuration, where 500 million minimum bias events of Xe+Cs interactions were collected (Table 4). As can be seen from the table, one can expect to observe the subthreshold production of $\Xi^-$ hyperon, while $\Lambda$ statistics will be sufficient to produce multidifferential distributions of physics variables as well as to measure, for instance, polarization of $\Lambda$ hyperons in heavy-ion interactions [37–39].

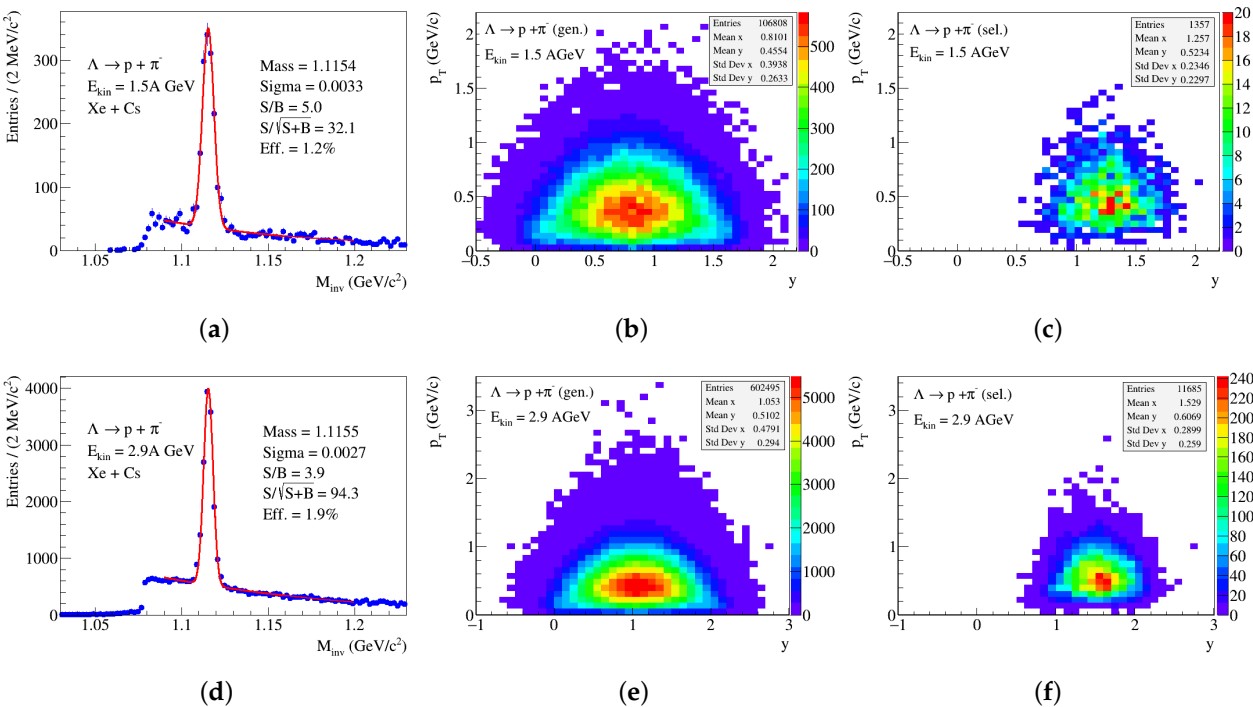

**Figure 10.** (**a**) Reconstructed invariant mass of $p$ and $\pi^-$ in $10^6$ minimum bias Xe+Cs interactions at Xe beam kinetic energy of 1.5 A GeV, (**b**) $y - p_T$ phase space of simulated $\Lambda$ hyperons, (**c**) phase space of hyperons after reconstruction and background suppression; (**d**–**f**)— the same as (**a**–**c**) for beam energy of 2.9 A GeV. Invariant mass distributions are fitted to a combination of a Gaussian and a polynomial function. Similarly to Figure 6, the legends include mean and sigma of the Gaussian, signal/background ratio, significance, and signal reconstruction efficiency. The signal and background are calculated within a $\pm 2\sigma$ interval of the peak position, and efficiency is taken with respect to all hyperons produced within 30 cm from the interaction point.

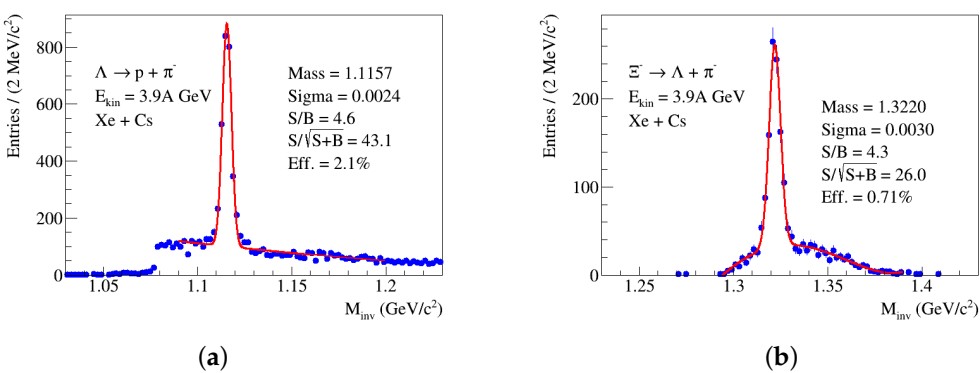

**Figure 11.** (**a**) Reconstructed invariant mass of $p$ and $\pi^-$ in $10^5$ minimum bias Xe+Cs interactions at the Xe beam kinetic energy of 3.9 A GeV, (**b**) reconstructed invariant mass of $\Lambda$ candidates and $\pi^-$ in $10^7$ events.

**Table 4.** $\Lambda$ and $\Xi^-$ hyperon production and reconstruction numbers for Xe+Cs interactions at 3.9 A GeV beam kinetic energy ($\sqrt{s_{NN}} = 3.296$ GeV): $E_{NN}^{thr}$ is the production threshold energy in $N + N$ interactions, $M$ is the hyperon production multiplicity per event, and $\epsilon$ is the reconstruction efficiency.

| Particle | $E_{NN}^{thr}$, GeV | $M$, Event$^{-1}$ | $\epsilon$, % | Yield / $5 \cdot 10^8$ Events |
|----------|---------|---------|---------|---------|
| $\Lambda$ | 1.6 | 1.06 | 2.0 | $1 \cdot 10^7$ |
| $\Xi^-$ | 3.7 | 0.012 | 0.7 | $4 \cdot 10^4$ |

## 5. Conclusions

Performance of the MPD and BM@N detectors at the NICA accelerator complex for hyperon reconstruction was presented for Monte Carlo simulations. The results demonstrate a good ability of both experiments to study strangeness production in heavy-ion interactions. It was also shown that advanced signal selection techniques based on ML approaches help to improve experimental performance for strange probes, resulting in the selection efficiency increase of $\sim$13, 29, and 33% for $\Lambda$, $\Xi^-$, and $\Omega^-$ hyperons, respectively, at the MPD setup.

**Author Contributions:** Conceptualization, V.K. (Vadim Kolesnikov) and A.Z.; methodology, M.K.; software, V.K. (Viktar Kireyeu); validation, D.S. and V.V.; investigation, A.M. and D.Z.; supervision, A.Z. All authors have read and agreed to the published version of the manuscript.

**Funding:** This research received no external funding.

**Institutional Review Board Statement:** Not applicable.

**Informed Consent Statement:** Not applicable.

**Data Availability Statement:** The data presented in this study are available on request from the corresponding author.

**Conflicts of Interest:** The authors declare no conflict of interest.

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
