# Peer review of "A Monte Carlo Study of Hyperon Production with the MPD and BM@N Experiments at NICA"

_2571-712X, doi:10.3390/particles6020027_

Round 1

Reviewer 1 Report

Overall, the paper is well written with a clear message. The paper has enough original materials presented. I have a few comments to be addressed by the authors:

Line 42: The MPD experiment at NICA is claimed to deliver a system size scan in the energy range 4-11 GeV. From Fig.1 it also follows that event rate has a strong dependence on collision energy. The statement given in the lines seems to be misguiding and does not quite describe the situation. Please consider to rephrase.

L69: It would have sense to reference ref.23 here for more details on the detector.

L101: I think that for true weak decays the distance dca_V0 should be minimal while in the text it is assumed that it should exceed some value. I guess it should be smaller than some value for true decay track combinations.

L131: The paper would benefit if authors presented a table with the list of selections (variable - value) optimized for reconstruction of each hyperon. Such a table would make it possible to compare MPD capabilities with other facilities where similar measurements are carried out.

Figure 7: MPD is a rather compact detector and the coverage should be dependent on the vertex position. Please specify how event vertex was distributed in the sampled events.

Figure 9: Statistical uncertainties of the measurements seem to be reasonably small, there is no significant pT dependence of uncertainties event for aL/L ratio. What is a reason that you stop your analysis at 3.5 GeV/c, why don't you look at higher momenta?

Figure 7 and 9: I see some contradiction between Fig.7 and Fig9. In Figure 7 you demonstrate that MPD does not have acceptance to measure lambdas at pT < 0.1-0.15 GeV/c at midrapidity. And yet in Fig.9 you present results for the lowest pT bin starting at zero momentum. This is misguiding and the analysis should have started from 0.1-0.15 GeV/c. It is probably too late to make the changes, but please take this comment into account in the future updates. Please also provide numbers in the text for the fraction of the total hyperon yields at midrapidity sampled by the detector, i.e. the ratios of generated spectra integrates in pt ranges [sampled range]/[whole range]. Such numbers would help to understand what fractions of the hyperon pT-integrated yields will have to be estimated from extrapolations.

Figure 10,11: The peak qualities for Lambda (and probably Xi) decays are much superior than those presented by MPD for real data measurements in the previous runs. Please provide details on the improvements in the detector and reconstruction procedures that resulted in such an impressive improvement.  Mentioning previous real data publications would be a good solution as well.

Reviewer 2 Report

  • L23: Is there a reference suitable for this statement?

  • L30: “...and to tighter constrain…” → “...and to improve constrains on th chemical freeze-out parameters.”

  • L49: “...offers the effective…” → “...offers an effective…”

  • L52-L53: Are there any TPC performance papers that can be referenced for the pT and dE/dx resolution?

  • L100: “...by applying different cuts (selection criteria)...” → “...by applying different selection criteria…”

  • L119: “The exact selection cut values can be found…” → “The exact selections can be found…”

  • L127: An explanation of the event mixing technique seems important here. Or perhaps a very brief description and a reference?

  • L204: “...Monte Carlo simulated data.” → “...Monte Carlo simulations.”

  • Conclusion section: I think the explicit addition of the improvement in terms of numbers would make the conclusion better.

Reviewer 3 Report

The paper describes MC studies of hyperons production with MPD and BM@N experiments at NICA facility. Overall, the paper is quite clear and the the method well explained. Nevertheless I suggest a significant revision of the language to simplify too-long sentences and clarify some of the procedures and methods described by the authors.

Considering that this study only relies on simulations, I'd  suggest to include a description of the simulation framework used to parametrize the detector response. This part is completely missing in the current version of the draft.

In case BM@A data form the first run  are already available (it is not clear from the paper if they are or not) I 'd suggest to include some experimental results (mass spectra, kinematic coverage, ...) and show how MC simulations compare.

In my opinion, with these additions and a careful check of the language, the paper would deserve to be published in Particles.

Here below some detailed comments for the authors.

1-7 for investigation of the objects with strangeness -> in strangness production.

1-18- probes-> processes

1-30-> put stringent constraints on the chemical ...

1-36 remove experimental

2-46 measurements ->measurement

2-51allows performing  the ..-> allows us to perform ... and to determine momenta ...

2-57 will make it possible to obtain  -> will provide

2-60 add a reference after 'shashlyk'

2-60 could you please specify what 'good resolution' means?

2-62 a high -> the high

2-63 registration -> detection

2-64 triggering  to the experiment with-> trigger with

2-68 could you  reword  to better explain 'event plane analysis'?

3-73 and 74 and 86downstream from -> downstream of

3-73 two-coordinate -> XY-coordinate GEM

3-89 characterized -with > determined by

3-89 Interaction trigger -> trigger

3-90with  trigger detectors BD and SiD->  with BD and SiD detectors.

4-94- found -> identifies

4-98 change 'approach'

4-1200 selection quality -> the selection is checked

4-101 'topological' i not clear -> replace with few words

4-101 the decay track combinations ' is not clear: could you please rephrase?

4-104 to 106 could you please rewrite it (what is distance path?) I'd check the whole paragraph. Srntences are too long with too many details . It would be good to simplify just focusing on the main feature of the secondary vertex detection.

4-122 replace combinations with something more specific (tracks?)

4-128 realized -> performed4-127 add a Ref. to event mixing technique.

4-127-19 could you please re-write the sentence to clarify it? what do you4

mean by multiple loops over selection criteria?

5-130 -> The final set of selection cuts was chosen based on the maximum...

5-143 it  should be reported how the detector response  was simulated: GEANT? parametric? FLUKA? this is an important information completely missing in the discussion.

5-145 remove particle combination

5-147 what does it mean 'for different event numbers from PHSD ...'? please rewrite and clarify

5-148-149 what does it mean 'reasonable selection efficiency'? please rephrase.

6-152 to 155 please rewrite the sentence: simplify and clarify.

7-165 do you mean  hyperon yield extracted by fitting the invariant mass spectra? please clarify and rephrase it. The procedure described here and in the following line is not clear to me.

8-173 to 17 Could you rephrase it breaking the long sentence in shorter one?

8-177 to 179: could you please better explain the procedure used? what do you mean by hyperon reconstruction w/o id?

8-186 it is not clear to me why for high energy collisions, the momentum resolution improves. Was the field different in the two configurations? or is it kept constant? please clarify.

8-195 were data collected or not yet? if already collected, why you are not showing the comparison with your MC simulations? otherwise the sentence needs to be rewritten to be more clear and precise.
